# Unfavorable Outcomes and Their Risk Factors in Hospitalized Patients with *Staphylococcus aureus* Bacteremia in the US: A Multicenter Retrospective Cohort Study, 2020–2022

**DOI:** 10.3390/antibiotics14030326

**Published:** 2025-03-20

**Authors:** Marya D. Zilberberg, Brian H. Nathanson, Rolf Wagenaar, Jan Posthumus, Andrew F. Shorr

**Affiliations:** 1EviMed Research Group, LLC, Goshen, MA 01032, USA; 2OptiStatim, LLC, Longmeadow, MA 01106, USA; brian.h.nathanson@att.net; 3Basilea Pharmaceutica International Ltd., 4123 Allschwil, Switzerland; rolf.wagenaar@basilea.com (R.W.); jan.posthumus@basilea.com (J.P.); 4Washington Hospital Center, Washington, DC 20010, USA; andrew.shorr@gmail.com

**Keywords:** *S. aureus*, hospitalization, outcomes, epidemiology

## Abstract

**Background**: In the US, 120,000 cases of *Staphylococcus aureus* bacteremia (SAB) occur annually. Apart from mortality, little is known about other unfavorable outcomes (UOs). We developed a multifaceted definition for UOs in SAB and examined their incidence and predictors. **Methods**: We conducted a multicenter (~300 hospitals) retrospective cohort study between 2020 and 2022 of adult hospitalized patients with at least one blood culture (BC) positive for *S. aureus*. UOs were any of the following: hospital mortality, antibiotic escalation, persistently positive BCs, prolonged post-infection length of stay (LOS), 30-day readmission, and disease worsening. We compared the group with UOs to favorable outcomes (FOs). Regression models identified predictors of UOs. **Results**: Among 4080 patients with SAB, 2427 (59.5%) experienced a UO, most commonly 30-day readmission (42.0%) and antibiotic escalation (37.7%). Those with UOs more frequently had septic shock at admission (5.7% vs. 1.2%), requiring the ICU (18.8% vs. 14.7%) and dialysis (4.4% vs. 1.9%) prior to SAB onset. Community-onset SAB predominated in both groups, with more complicated SAB in the UO group (39.8% vs. 22.3%). Vancomycin use was similar, while daptomycin was more common in the UO group (8.5% vs. 3.0%). Variables with the highest odds ratios predicting a UO were septic shock on admission (3.498, 95% CI 2.145, 5.704), empiric daptomycin (2.723, 95% CI 1.943, 3.821), and complicated SAB (2.476, 95% CI 2.047, 2.994). **Conclusions**: UOs occur frequently in the setting of SAB. A broader perspective exploring issues other than mortality demonstrates the substantial implications of SAB both for patients and healthcare systems. Select clinical variables are associated with UOs, some of which may not be modifiable.

## 1. Introduction

*Staphylococcus aureus* bacteremia (SAB) remains a common problem both in the US and worldwide, with an annual incidence of ~20/100,000 population, adding up to 120,000 cases [1,2]. Generally classified as either primary or secondary (associated with a device or a distant infection locus) in nature, SAB can also occur as either a complicated or uncomplicated infection. SAB, moreover, may arise as either community acquired or healthcare associated. Adding further complexity, hospital-acquired SAB is commonly caused by community-acquired variants. Such complexity makes clinical management challenging.

While diminishing in prevalence over the recent decades, methicillin-resistant *S. aureus* (MRSA) continues to comprise nearly one-half of all *S. aureus* isolates in the US [2]. Because of its high prevalence, patients suspected of suffering from SAB are routinely empirically treated with anti-MRSA agents, such as vancomycin and daptomycin. Despite the availability of such effective therapies, whether for methicillin-sensitive (MSSA) or MRSA isolates, mortality in SAB has plateaued in the US at 20% [2,3].

Although risk factors for certain outcomes are known and diagnostic and treatment paradigms exist, there remains a dearth of evidence-based guidelines for the comprehensive management of SAB [4]. Few studies have examined important outcomes other than mortality in SAB, such as persistent infection, complications, and/or readmission, outcomes that may inform strategies for preventing these complications through intensive interventions among patients at risk.

In the current retrospective cohort study, we sought to describe SAB patients who are prone to various adverse outcomes and to determine whether they can be identified early in the course of SAB and distinguished from the group who are likely to do well.

## 2. Methods

### 2.1. Ethics Statement

Because this study used fully de-identified administrative data, it was exempt from ethics review under US 45 CFR 46.101(b)4 [5]. All methods were carried out in accordance with relevant guidelines and regulations. No ethics review was sought due to the guidance cited above [5].

### 2.2. Study Design and Patient Population

We conducted a multicenter retrospective cohort study of hospitalized patients with documented SAB in the US. We included MRSA and methicillin-sensitive *S. aureus* (MSSA) cases, as well as complicated and uncomplicated infections. We further evaluated patients with community-acquired (CA) and hospital-acquired (HA) SAB (for full definitions, please see Appendix A). The case identification approach relied on ICD-10 codes combined with microbiology results and pharmacy data. We used microbiology data to identify the primary source of infection. The onset of infection was defined as the day the index culture was obtained. SAB present within the first 3 days of admission was classified as community acquired. Microbiology and pharmacy data were used to determine the timeliness and appropriateness of treatment.

Patients were included if they were adult (age ≥ 18 years) inpatients with at least one blood culture during the index hospitalization positive for *S. aureus* and if they received one of the following intravenous (IV) anti-*S. aureus* agents for at least 3 days: vancomycin, daptomycin, ceftaroline, cefazolin, oxacillin, or nafcillin. Only the first SAB episode within the study time frame was included. Those <18 years of age, those without a blood culture positive for *S. aureus*, persons with a polymicrobial bloodstream infection, and those transferred from another acute care facility were excluded.

### 2.3. Data Source

The data for the study derived from the PINC AI Healthcare Database (formerly Premier Healthcare Database), an electronic laboratory, pharmacy, and billing data repository, for the years 2020 through 2022. The database contains inpatient records of 13 million hospital discharges annually, representing approximately 25% of all hospitalizations nationwide in nearly 900 hospitals. In addition to the standard information contained in typical hospital claims, such as patient age, gender, race/ethnicity, principal and secondary diagnoses, and procedures, the database contains a date-stamped log of all items and services charged to the patient or their insurer, including all medications, laboratory tests, and diagnostic and therapeutic services. Premier assigns each patient a unique identifier so that previous and subsequent admissions to the same Premier hospital, along with principal and secondary diagnoses and procedure codes, can be readily ascertained. For the current study, we used a subset of approximately 300 US institutions that submit microbiology data into PINC AI. Eligible time began only following the commencement of microbiology data submission by each institution. For further descriptions of the database, please see references [6,7,8,9,10].

### 2.4. Baseline Measures

Baseline measures included both hospital (geographic area, size, urbanicity, and academic affiliation) and patient (demographics and clinical) characteristics. In addition to standard factors, we evaluated their exposure to antibiotics during the 90 days prior to the index admission. We computed a Charlson comorbidity score as a measure of the burden of chronic illness.

### 2.5. Pre-Infection Onset Hospital and Process-of-Care Variables

Several hospital-based pre-infection onset characteristics and process-of-care variables were examined. These included multiple illness severity markers: (1) ICU and time to ICU admission relative to the date of hospitalization (if before index infection) among those requiring any ICU care; (2) endotracheal intubation/mechanical ventilation (MV) and time to MV relative to the date of hospitalization (if before index infection) among those requiring any MV (ICD-10 code 5A1945Z or 5A1955Z); (3) vasopressor use and time to vasopressor administration relative to the date of hospitalization (if before index infection) among those requiring any vasopressors; and (4) dialysis and time to dialysis relative to the date of hospitalization (if before index infection) among those requiring any dialysis (ICD-10 codes 5A1D00Z, 5A1D60Z, 3E1M39Z, or CPT code 90935). Additionally, antibiotic exposure during the index hospitalization prior to the onset of the infection was recorded. We established the prevalence of severe sepsis (ICD-10 code R6520) or septic shock (ICD-10 code R6521) at any time during index hospitalization.

### 2.6. Infection and Treatment Characteristics

Organisms and their susceptibilities were identified from the blood culture that defined the onset of SAB. All microbiologic testing was conducted by local laboratories. Empiric antibiotic treatment was considered appropriate and timely if the patients received vancomycin (at least ~15 mg/kg IV every 12 h, equating to 500 mg IV very 6 h or 1 gm IV every 12 h) or daptomycin (at least ~6–8 mg/kg IV daily, equating to 350 mg to 1 gm IV once per day) if MRSA or cefazolin (2 g IV every 8 h), oxacillin (2 g IV every 4 h), or nafcillin (0.5–2 g IV every 4 h or 3 g IV every 6 h) if MSSA within 2 days of the index culture being obtained. All other antibiotic regimens were termed inappropriate empiric treatment (IET). A full treatment course was considered to be 14 days for uncomplicated and 28 days for complicated SAB. For the patients discharged alive prior to completing their IV treatment, evidence of the continuation of IV treatment through the day of discharge categorized them into the appropriate duration. In addition, we examined early (within 2 days of index culture) vs. late (day 3 or later post-index culture) source controls [11].

### 2.7. Outcome Variables

Based on clinical importance to patients, clinicians, and institutions, a composite variable “unfavorable outcome” (UO) served as the primary endpoint. The components of this variable were as follows:Death during index hospitalization (hospital mortality).Worsening of disease:
Antimicrobial escalation:
Switch from vancomycin to daptomycin;Switch from vancomycin to ceftaroline;Switch from daptomycin to ceftaroline;Switch from vancomycin to daptomycin + ceftaroline;Addition of ceftaroline to daptomycin;Addition of an aminoglycoside or a fluoroquinolone to vancomycin;Addition of an aminoglycoside or a fluoroquinolone to daptomycin.
ICU following SAB onset;MV following SAB onset;Vasopressors following SAB onset;Dialysis following SAB onset.


3.Failure to eradicate *S. aureus* (SA, defined as blood cultures on day 8 or later following index culture continuing to grow index organism).4.Prolonged post-infection onset length of stay (LOS, defined as longer than the group median LOS).5.Readmission for any reason within 30 days of discharge among survivors.

Secondary outcomes included each component of the composite primary outcome, as well as the following:Post-infection onset hospital LOS;Post-infection onset ICU LOS (days);Duration of post-infection onset MV;Post-infection onset hospital costs ($).

### 2.8. Follow-Up

The cohort was followed longitudinally until discharge or death in the hospital. Survivors were followed for additional 30 days for the outcome of 30-day readmission.

### 2.9. Statistical Analyses

We used standard descriptive statistics to compare patients with UOs to those achieving FOs across demographics, comorbidities, hospital characteristics and processes, as well as hospital outcomes. Continuous variables are reported as means with standard deviations (SDs) when normally distributed and/or as medians with 25th and 75th percentiles when the distributions are skewed. Differences between the mean values were tested via Student’s *t*-test, while those between medians were examined using the Mann–Whitney U test. Categorical data are summarized as proportions. The Chi-square test or Fisher’s exact test for cell counts <4 was used to examine inter-group differences. We developed a predictive model for UOs based on the baseline, infection, and early hospital treatment characteristics. The measure of discrimination was the area under the receiver operating characteristics curve (AUROC), while calibration was tested using the Hosmer–Lemeshow goodness-of-fit test. Statistical significance was set at *p* < 0.05.

## 3. Results

Between the years 2020 and 2022, 4080 patients with SAB met the enrollment criteria, of whom 2427 (59.5%) suffered a UO. Compared to those with FOs, patients with UOs were more likely to be non-White and suffered from a higher comorbidity burden (median [IQR] Charlson score of 3 [1, 5] in UOs vs. 2 [1, 4] in FOs, *p* < 0.001, Table 1; Appendix A for individual comorbidities). In both groups, the largest hospitals (500+ beds) housed more patients overall, and the prevalence of patients with UOs was directly, and of those with FOs, inversely, proportional to hospital size. In addition, patients with UOs were more likely than those with FOs to be admitted to urban and teaching hospitals (Table 1).

During the hospitalization and prior to the onset of SAB, those with UOs were more severely ill, as demonstrated by a higher prevalence of such measures of acute illness as early septic shock, the need for an ICU, MV, vasopressor support, and dialysis (Table 2). Additionally, patients with UOs had the onset of at least some of these markers (e.g., time to MV) occur later during hospitalization, but their duration was similar to that among patients with FOs. For example, the need for MV trended toward more frequent in the UO group than in the FO group, and the mean (SD) time to MV was 3.0 (3.9) days among the UO group vs. 2.0 (2.3) days in FO population (*p* = 0.052). However, the mean (SD) MV duration in the UO group [6.0 (8.8) days] was similar to that in the FO group [5.7 (8.9) days] (*p* = 0.846). Septic shock was present on admission nearly five times more frequently in the setting of a UO than an FO (Table 2).

The infection characteristics are presented in Table 3. Although MSSA predominated in both groups, SAB leading to UOs was more likely to be caused by MRSA (44.8% vs. 33.7%, respectively, *p* < 0.001). Similarly, compared to that in the FO group, infection in the UO group was more often complicated (39.8% vs. 22.3%, *p* < 0.001), with the two most common measures being persistent bacteremia and secondary bloodstream infection (BSI). There was no difference between the groups with respect to whether the SAB was community or hospital acquired. Vancomycin and cefazolin were the most common antimicrobials utilized empirically in both groups. Empiric daptomycin was rare, although nearly three times more common in the UO group. Although IET was more common in the group with FOs than UOs (3.3% vs. 2.0%, *p* = 0.007), the rates were low in both groups. In both groups, few had evidence of a source control procedure. Most patients (80.6% in UO group and 77.3% in FO group, *p* < 0.013) completed their definitive treatment course (Table 3).

The most prevalent UOs were 30-day readmission (42.0%) and antibiotic escalation (37.7%, most frequently a switch from vancomycin to daptomycin; Appendix A), while mortality was an uncommon UO (5.1%) (Figure 1). The group with UOs had a longer post-infection onset hospital LOS and higher post-infection onset hospital costs (Table 3). In fact, the mean ± SD post-infection onset hospital costs in the setting of a UO were double those with FOs ($37,800 ± $56,910 vs. $18,371 ± $11,249, *p* < 0.001). In a predictive model, the variables with the highest odds ratios predicting a UO were the presence of septic shock on admission, empiric treatment with daptomycin, and SAB that was complicated (Table 4). The model had moderate discrimination and calibration.

## 4. Discussion

In the current study, we demonstrate that a substantial proportion of patients hospitalized with SAB suffer many unfavorable outcomes despite the vast majority receiving timely and appropriate initial therapy. In particular, survivors of SAB with UOs were subject to a staggering 42.0% 30-day rehospitalization rate. And even across the entire cohort, inclusive of both UOs and FOs, nearly 1 in 4 survivors (23.7%) required readmission within 30 days of discharge. This is substantially higher than the 13.9% average rate of 30-day readmission to acute care US hospitals overall and approximately 50% higher relative to the 15.6% rate reported for infectious and parasitic diseases in general [12]. In SAB specifically, an analysis by Inagaki and colleagues based on 2014 data noted a 22% overall 30-day readmission rate, which is nearly identical to our overall 30-day readmission rate across UOs and FOs combined [13]. The same study reported hospital mortality in 13% of subjects in contrast to our cohort, whose mortality was far lower at 5.1% among UOs, and a mere 3% among the entire cohort. These discrepancies are likely due to vastly different data sources and case definitions. That is, Inagaki et al. relied on administrative coding within the National Readmission Database from the Agency for Healthcare Research and Quality, a source with limited data points for each patient. Additionally, using administrative codes alone, rather than the results from actual cultures as we did, to identify cases of SAB introduces a high risk for misclassification [14]. In contrast, our data came from an electronic medical record database rich in patient variables, including microbiology, which we used to confirm *S. aureus* bacteremia, thus limiting this attendant risk. A recent systematic review of studies reporting the mortality for patients with SAB found that the one-month mortality rate ranged from 6% to 35% in analyses performed in 2011 or later, with a mean of 20% [15]. Since the median hospital LOS in our study was 12 days, it is expected that our hospital mortality tracks closer to the lower bound of the number of events at one month. It is possible that the patterns of these outcomes have, in fact, shifted over the span of the near decade between their study and ours. Further studies should clarify what accounts for such discrepancies across time and subpopulations of patients with SAB.

Other unfavorable outcomes abound among patients with SAB. For example, over one-third of those with UOs required antimicrobial escalation during their hospitalization. Hand in hand with that, one-third required some period of vasopressor support, while one in ten needed dialysis or admission to the ICU. Importantly, in 20% of patients with a UO, the hospital LOS was longer than the median LOS among SAB patients more generally. While each of the above may not be as clinically or humanistically devastating as death, together they represent a substantial morbidity burden both to patients and to the healthcare system. Without first understanding the burden of these events and then identifying the patients at risk for them, it is difficult to improve outcomes.

For these reasons, we asked an additional question: are there baseline or treatment variables that predict UOs in patients with SAB? Indeed, we have identified several. While suffering from multiple comorbidities significantly increases the risk of a UO, the strongest predictors of a UO were more related to infection characteristics and treatment patterns. Not surprisingly, septic shock, when present on admission, portended a poor outcome, as did complicated SAB and SAB with MRSA rather than MSSA.

The only potentially modifiable UO risk factors were the empiric treatment choices of daptomycin and vancomycin. On the one hand, both may be a byproduct of confounding by indication, whereby patients receiving these drugs are sicker than those receiving other therapies, but their level of illness is lost in the residual confounding. On the other, and particularly given the magnitude of their effect, this finding may be presenting a need for new therapeutic approaches that could improve outcomes in SAB.

Interestingly, an age of 85 years or older appeared to be protective against UOs. The study by Inagaki and colleagues, which examined 30-day readmissions along with hospital mortality, LOS, and costs among patients with SAB, noted higher readmission rates in the younger groups than among patients 80 years or older, although this relationship did not hold up upon adjusting for confounders, and it was not examined in the context of the remaining outcomes [13]. Age has been reported to be the most important predictor of another unfavorable outcome: mortality. In a broad literature review on this topic, van Hal et al. examined over 40 studies that addressed this issue [16]. Overall, advanced age was associated with a higher risk of mortality. However, mortality in all of those studies was much higher than what we have observed in our cohort, and patients 85 years and older were not considered as a single category. A more recent effort attempted to define several relevant phenotypes of SAB patients [4]. Drawing on data from three large, separate studies, the investigators defined five unique groups based on various common factors their members possessed. Group A, defined by older age, comorbidity, and SAB from an unknown source or skin or soft tissue infection, exhibited the highest mortality. In contrast to our study, Swets and colleagues did not examine the most elderly within the old group. Additionally, there were many differences in the cohort composition, one of the most striking ones being age distribution. That is, in the three studies comprising the phenotyping analysis, the median age in each group was 65 years or older, whereas in ours, the median age was lower. Another big difference is the location of the studies, wherein all three of the cohorts in Swets’s analyses were outside the US, and ours was limited to US institutions, thus raising the question of whether geographic differences in healthcare delivery played a role [17]. Finally, to the best of our knowledge, we are the first group to focus on a broader swath of relevant unfavorable outcomes in this population of patients, thus raising questions for future investigation. One hypothesis for this finding may be that patients 85 years and older may not be subjected to such aggressive interventions as dialysis, ICU, MV, or readmission. In this way, this lack of aggressive treatment we defined as a UO may be masking the withdrawal of aggressive care rather than a true FO. Future studies should address this apparent paradox in further detail.

One important feature of our study is that it includes data from the initial year of the COVID-19 pandemic in the US. Although we did not analyze that year separately, it is generally reported that patients suffering from COVID-19 appear to be more prone to nosocomial SAB with MRSA, and the coexistence of both leads to higher mortality [18,19]. Because the vast majority of our cohort had community-acquired SAB and because COVID-19 infection was seen in a small minority of patients without clear association with UOs, we did not examine its potential influence as an effect modifier of SAB outcomes separately. In addition, evidence that the quality of SAB treatment and its outcomes are not altered by COVID-19 further supports our approach [20].

Our study has a number of strengths and limitations. By virtue of it being a retrospective cohort study, it is prone to several forms of bias, most notably selection bias. We mitigated this by establishing a priori enrollment criteria and enrolled all consecutive patients that met them. Confounding is a risk as well in any observational study. We did not specifically address this issue when comparing patients with UOs and FOs because, by definition, their outcomes are confounded by disease severity and other factors. However, in our model predicting the occurrence of UOs, we did identify several associated covariates. As we stated above, the antimicrobial treatment variables retained in the model may not, by themselves, drive the outcomes but rather may be markers of more severe disease that are not captured in the data. This potentially makes them subject to confounding by indication, a form of residual confounding. The magnitude of this effect, in turn, makes residual confounding as the only explanation for this finding unlikely. Misclassification was reduced by choosing a cohort of patients with confirmed *S. aureus* bacteremia, and specificity was optimized by requiring antimicrobial treatment. While some of the outcomes may be more prone to misclassification (e.g., persistent infection), others, such as hospital mortality, LOS, and costs, are not. One potential source of misclassification is that we did not impose a time frame on treatment escalation. At the same time, the median time to escalation was 6 days, in contrast to the median LOS of 12 days, making a switch for reasons other than escalation unlikely. Finally, the nature of the database did not allow us to explore the potential impact of a consultation from an Infectious Diseases specialist, which has been reported to be associated with improved outcomes in SAB [21]. The biggest strength of our analysis is its generalizability, given that it is a large multicenter cohort within representative US institutions.

## 5. Conclusions

In summary, among hospitalized patients with SAB, the vast majority of it being community associated, despite a relatively low hospital mortality rate, the combined prevalence of untoward outcomes is over 70%. Notably, the most common UO is 30-day readmission, which occurs in over 40% of all survivors. While most of the baseline and early treatment risk factors for a UO are not modifiable, recognizing their presence may help clinicians identify patients who may benefit from more aggressive measures to prevent complications. Furthermore, while empiric treatment with daptomycin and vancomycin, as identified in our model as a significant risk factor, may be, at least in part, due to confounding by indication, the magnitude of the effect suggests that there is ample room for novel treatments to explore reducing the risk of these common morbidities among patients with SAB.

## Figures and Tables

**Figure 1 antibiotics-14-00326-f001:**
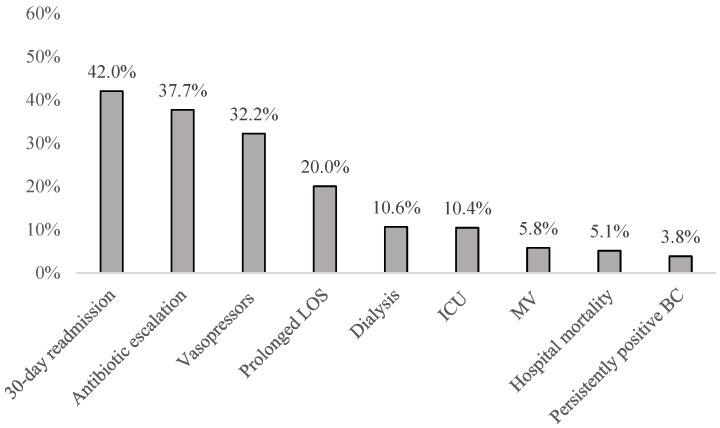
Distribution of individual unfavorable outcomes. Unfavorable outcome prevalence among patients with at least one unfavorable outcome; all outcomes are post-infection onset. LOS = length of stay; ICU = intensive care unit; MV = mechanical ventilation; BC = blood culture.

**Table 1 antibiotics-14-00326-t001:** Baseline characteristics.

	Favorable Outcomes	%	Unfavorable Outcomes	%	*p*-Value
	N = 1653		N = 2427		
Admission year					
2020	616	37.27%	840	34.61%	0.179
2021	597	36.12%	934	38.48%	
2022	440	26.62%	653	26.91%	
Age, years					
Mean (SD)	60.8 (17.0)		60.0 (16.3)		0.098
Median [IQR]	62 [50, 74]		61 [49, 72]		0.067
Gender: male	1060	64.13%	1536	63.29%	0.585
Race					
White	1384	83.73%	1896	78.12%	<0.001
Black	164	9.92%	332	13.68%	
Asian	16	0.97%	30	1.24%	
Other	72	4.36%	131	5.40%	
Unknown	17	1.03%	38	1.57%	
Hispanic ethnicity	101	6.11%	195	8.03%	0.020
Admission source					
Non-healthcare facility (including from home)	1505	91.05%	2191	90.28%	0.490
Clinic	64	3.87%	83	3.42%	
Transfer from ECF or ICF	47	2.84%	90	3.71%	
Transfer from another non-acute care facility	29	1.75%	51	2.10%	
Other	8	0.48%	12	0.49%	
Admission type					
Emergency	1508	91.23%	2209	91.02%	
Urgent	92	5.57%	146	6.02%	
Elective	39	2.36%	54	2.22%	
Other/information not available	14	0.85%	18	0.74%	0.905
Do Not Resuscitate order					
Present on admission	129	7.80%	174	7.17%	0.448
Any time	182	11.01%	321	13.23%	0.035
Charlson comorbidity score					
0	338	20.45%	377	15.53%	<0.001
1	318	19.24%	369	15.20%	
2	290	17.54%	384	15.82%	
3	191	11.55%	315	12.98%	
4	174	10.53%	311	12.81%	
5+	342	20.69%	671	27.65%	
Mean (SD)	2.6 (2.3)		3.1 (2.4)		<0.001
Median [IQR]	2 [1, 4]		3 [1, 5]		<0.001
Hospital characteristics					
Census region					
Midwest	299	18.09%	477	19.65%	0.145
Northeast	353	21.36%	486	20.02%	
South	968	58.56%	1433	59.04%	
West	33	2.00%	31	1.28%	
Number of beds					
<100	144	8.71%	120	4.94%	<0.001
100 to 199	283	17.12%	308	12.69%	
200 to 299	262	15.85%	318	13.10%	
300 to 399	195	11.80%	277	11.41%	
400 to 499	206	12.46%	362	14.92%	
500+	563	34.06%	1042	42.93%	
Teaching	807	48.82%	1414	58.26%	<0.001
Urban	1318	79.73%	2049	84.43%	<0.001

SD = standard deviation; ECF = extended care facility; ICF = intermediate care facility; IQR = interquartile range.

**Table 2 antibiotics-14-00326-t002:** Pre-infection onset hospital markers of acute illness severity.

	Favorable Outcomes	%	Unfavorable Outcomes	%	*p*-Value
	N = 1653		N = 2427		
**ICU admission**	243	14.70%	457	18.83%	0.001
Time from hospital admission to ICU admission, days					
Mean (SD)	1.5 (1.6)		1.6 (2.3)		0.248
Median [IQR]	1 [1, 1]		1 [1, 1]		0.467
ICU LOS, days					
Mean (SD)	3.3 (4.9)		3.4 (6.3)		0.740
Median [IQR]	1 [1, 4]		1 [1, 3]		0.482
**MV**	75	4.54%	143	5.89%	0.059
Time from hospital admission to MV, days					
Mean (SD)	2.0 (2.3)		3.0 (3.9)		0.052
Median [IQR]	1 [1, 2]		1 [1, 3]		0.081
MV duration, days					
Mean (SD)	5.7 (8.9)		6.0 (8.8)		0.846
Median [IQR]	2 [1, 6]		2 [1, 7]		0.566
**Septic shock POA**	20	1.21%	138	5.69%	<0.001
**Vasopressors**	110	6.65%	184	7.58%	0.261
Time from hospital admission to vasopressors, days					
Mean (SD)	2.5 (2.9)		3.0 (4.5)		0.338
Median [IQR]	1 [1, 2]		1 [1, 3]		0.424
Vasopressors duration, days					
Mean (SD)	1.6 (1.6)		1.5 (1.2)		0.568
Median [IQR]	1 [1, 1]		1 [1, 1]		0.831
**Dialysis**	31	1.88%	107	4.41%	<0.001
Time from hospital admission to dialysis, days					
Mean (SD)	2.4 (3.0)		2.0 (2.9)		0.498
Median [IQR]	1 [1, 2]		1 [1, 2]		0.611
Dialysis duration, days					
Mean (SD)	1.7 (1.6)		2.2 (3.2)		0.406
Median [IQR]	1 [1, 2]		1 [1, 2]		0.267

ICU = intensive care unit; SD = standard deviation; IQR = interquartile range; MV = mechanical ventilation; POA = present on admission.

**Table 3 antibiotics-14-00326-t003:** Infection characteristics and outcomes.

	Favorable Outcomes	%	Unfavorable Outcomes	%	*p*-Value
	N = 1653		N = 2427		
**Co-incident or co-prevalent COVID-19**	169	10.22%	293	12.07%	0.067
**Organism**					
MRSA	557	33.70%	1087	44.79%	<0.001
MSSA	1074	64.97%	1325	54.59%	
Unknown	22	1.33%	15	0.62%	
**Complicated BSI**	369	22.32%	966	39.80%	<0.001
Persistent bacteremia	179	10.83%	436	17.96%	<0.001
TEE	13	0.79%	30	1.24%	0.167
H/o hemodialysis	31	1.88%	326	13.43%	<0.001
Secondary BSI	189	11.43%	395	16.28%	<0.001
SSTI	8	0.48%	14	0.58%	0.691
Joint	22	1.33%	49	2.02%	0.099
Bone	26	1.57%	56	2.31%	0.101
Vascular	8	0.48%	14	0.58%	0.691
CSF	0	0.00%	5	0.21%	0.085
Other CNS	0	0.00%	1	0.04%	1.000
Heart	55	3.33%	92	3.79%	0.436
Lung	33	2.00%	70	2.88%	0.076
Pleura	8	0.48%	14	0.58%	0.691
CLABSI	40	2.42%	98	4.04%	0.005
**Primary BSI**	1464	88.57%	2032	83.72%	<0.001
**Acquisition location**					
Community acquired	1416	85.66%	2074	85.46%	0.853
Hospital acquired	237	14.34%	353	14.54%	
**Treatment by day 2 of infection onset**					
Drug					
Vancomycin	1413	85.48%	2152	88.67%	0.003
Daptomycin	50	3.02%	205	8.45%	<0.001
Ceftaroline	6	0.36%	105	4.33%	<0.001
Cefazolin	577	34.91%	756	31.15%	0.012
Oxacillin	60	3.63%	76	3.13%	0.384
Nafcillin	63	3.81%	121	4.99%	0.076
Other (none of the above)	64	3.87%	65	2.68%	0.032
IET	55	3.33%	48	1.98%	0.007
**Antimicrobial course completed**	1332	80.58%	1877	77.34%	0.013
**Evidence of source control procedure**	16	0.97%	40	1.65%	0.067
**Post-infection onset outcomes**					
ICU LOS, days					
Mean (SD)			5.1 (6.2)		
Median [IQR]			3 [2, 6]		
Hospital LOS, days					
Mean (SD)	9.2 (3.7)		15.5 (14.2)		<0.001
Median [IQR]	8 [7, 11]		12 [8, 18]		<0.001
Hospital costs, $					
Mean (SD)	18,371 (11,249)		37,800 (56,910)		<0.001
Median [IQR]	15,682 [11,125, 22,182]		26,546 [16,698, 43,124]		<0.001

MRSA = methicillin-resistant *S. aureus*; MSSA = methicillin-sensitive *S. aureus*; BSI = bloodstream infection; TEE = transesophageal echocardiogram; h/o = history of; SSTI = skin or soft tissue infection; CSF = cerebrospinal fluid; CNS = central nervous system; CLABSI = central line-associated bloodstream infection; IET = inappropriate empiric therapy; ICU = intensive care unit; LOS = length of stay.

**Table 4 antibiotics-14-00326-t004:** Predictors of unfavorable outcomes.

Covariate	Odds Ratio	95% CI	*p* > |z|
*Demographics*			
Age ≥ 80	0.687	(0.564 to 0.836)	<0.001
*Comorbidities*			
Charlson Index (per unit)	1.056	(1.026 to 1.087)	<0.001
Weight loss	1.658	(1.349 to 2.037)	<0.001
Deficiency anemias	1.499	(1.306 to 1.722)	<0.001
Valvular disease	1.322	(1.122 to 1.559)	0.001
*Infection Characteristics*			
Septic shock present on admission	3.498	(2.145 to 5.704)	<0.001
Complicated BSI	2.476	(2.047 to 2.994)	<0.001
MRSA (vs. MSSA or unknown)	1.473	(1.281 to 1.694)	<0.001
Secondary BSI	0.704	(0.549 to 0.904)	0.006
*Treatment by day 2 of infection onset*			
Daptomycin	2.725	(1.943 to 3.821)	<0.001
Vancomycin	1.451	(1.179 to 1.786)	<0.001

AUROC = 0.672 (95% CI 0.656, 0.689); Hosmer–Lemeshow *p* = 0.107. CI = confidence interval; BSI = bloodstream infection; MRSA = methicillin-resistant *S. aureus*; MSSA = methicillin-susceptible *S. aureus.*

## Data Availability

The dataset analyzed in the current study was used under a license from PINC AI and is available from the corresponding author on reasonable request and with permission from PINC AI. A portion of these data have been accepted for presentation at the IDWeek annual meeting 2024.

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
