# Peer review of "Unfavorable Outcomes and Their Risk Factors in Hospitalized Patients with Staphylococcus aureus Bacteremia in the US: A Multicenter Retrospective Cohort Study, 2020–2022"

_antibiotics, 2025, doi:10.3390/antibiotics14030326_

Round 1

Reviewer 1 Report

Comments and Suggestions for Authors

The manuscript describes a retrospective study conducted across 300 hospitals in US with an aim to identify the unfavourable outcomes and associated risk factors in patients diagnosed with Staphylococcus bacteraemia. The paper has been written very clearly, the methodology is explicitly explained and aligns with the study objectives. Few minor queries from my side are:

Line no. 158 to 162: case definitions for the listed secondary objectives should be added for better understanding of the readers.

The study identified the choice of empirical antibiotic treatment (vancomycin/ daptomycin) as one of the significant modifiable risk factors for UOs. This, as the authors mention may partly be attributable to confounding by indication.

  • Which particular UOs were seen in patients receiving the two antibiotics in question?
  • What were the results with other antibiotics given by day 2 of infection onset?
  • Were the patients on single or more than 1 antibiotics?  

Author Response

The manuscript describes a retrospective study conducted across 300 hospitals in US with an aim to identify the unfavourable outcomes and associated risk factors in patients diagnosed with Staphylococcus bacteraemia. The paper has been written very clearly, the methodology is explicitly explained and aligns with the study objectives. Few minor queries from my side are:

Line no. 158 to 162: case definitions for the listed secondary objectives should be added for better understanding of the readers.

The study identified the choice of empirical antibiotic treatment (vancomycin/ daptomycin) as one of the significant modifiable risk factors for UOs. This, as the authors mention may partly be attributable to confounding by indication.

  • Which particular UOs were seen in patients receiving the two antibiotics in question?

AU: The reviewer asks an important question. Since the aim of this analysis was to establish the risk factors for the composite outcome “unfavorable outcomes,” we did not examine the specific contribution of each of the covariates on each of the components of UO. However, we agree that, given our findings, the next step may be to ask this question, if the numbers allow for a meaningful statistical analysis.

  • What were the results with other antibiotics given by day 2 of infection onset?

AU: The other antibiotics did not produce a signal with regard to increasing the risk of a UO.

  • Were the patients on single or more than 1 antibiotics?

AU: We did not exclude patients who were on multiple antibiotics. So, yes, they could have been on more than one.   

Reviewer 2 Report

Comments and Suggestions for Authors

Dear Editor,

Thank you for your invitation to revise the manuscrit entitled 'Unfavorable outcomes and their risk factors in hospitalized patients with Staphylococcus aureus bacteremia in the US: A mul- 3
ticenter retrospective cohort study, 2020-2022'. It is a well written manuscript and has high degree of academic significance. Staphylococcus aureus is an important infectious disease that can be mortal and severe, so the management is particularly important. The manuscript can be accepted.

Author Response

Dear Editor,

Thank you for your invitation to revise the manuscrit entitled 'Unfavorable outcomes and their risk factors in hospitalized patients with Staphylococcus aureus bacteremia in the US: A mul- 3
ticenter retrospective cohort study, 2020-2022'. It is a well written manuscript and has high degree of academic significance. Staphylococcus aureus is an important infectious disease that can be mortal and severe, so the management is particularly important. The manuscript can be accepted.

AU: We thank the reviewer for their feedback.

Reviewer 3 Report

Comments and Suggestions for Authors

Appreciate the opportunity to review this manuscript on SAB and unfavorable outcomes drawn from a large sample out of a large database.  I enjoyed reading the manuscript.  Appreciate the supplemental material, tables and the figure which I am sure required extensive work.  As I read the manuscript, I found some areas in which clarification will be necessary and could further strengthen you work.  The English language and terminology will require minor editing.  Few inconsistencies among abbreviations.  I hope my comments will be well received and easy to follow. 

Good luck

--------------

MANUSCRIPT ID: antibiotics-3497311-v1

TITLE: Unfavorable outcomes and their risk factors in hospitalized patients with Staphylococcus aureus bacteremia in the US: A multicenter retrospective cohort study, 2020 -2022

GENERL COMMENTS: Thank you for your efforts on preparing this interesting manuscript.  I really enjoyed reading and even learning/refreshing a few terms during my review.  The manuscript is well written and for most parts flows well.  However, due to the multiple/many abbreviations and the repetitive nature (i.e., SAB, UO, FO), it was difficult to follow at some sections.  SAB is an infection, so at some parts you may just refer to it as such.  Few terminologies although correct, do not necessarily fit in the context and I will suggest revising them for better flow.  Multiple inconsistencies with the abbreviations, UO, FO, SAB, etc.   For most part, I pointed them out for you in specific lines. 

You have used “Unfavorable outcomes” (UOs) and untoward outcomes.  I think you should keep the terminology consistent. 

Not sure if you need to have the time frame (2020 – 2022) listed in the title.  Title is already long.  In addition, may consider reversing the order of the title: Risk factors associated with UOs in hospitalized patients with SAB in the US: A multicenter retrospective cohort study.

ABSTRACT:

Line #16 – Since you have outcomes, the (UO) should be (UOs).  In addition, I think “incidence” or “frequency” may fit better than “prevalence”.

Line #17 – Not sure if your time frame 2020 – 2022 belongs here since we are discussing centers.  It can easily go to the end of the sentence (i.e., for S. aureus between 2020- 2022).

Line #18 – you have 1 + BC positive for S. aureus.  Are you trying to say with one positive S. aureus blood culture?  Please clarify you have (+) and positive.  Do you mean more than one positive culture?

Line #20 – Should you abbreviate (LOS) here since you have other abbreviations in the abstract. 

Line #21 – Add “and” before disease worsening.  Complete the term before abbreviating.  Favorable outcomes (FOs).  Then, we compared the UO group to favorable outcome (FO) group. 

Line #22 – I think “experienced” or “encountered” may be a better term than “suffered”.

Line #25 – May consider changing to: required ICU admission….

Line #30 – Just a minor change in this short sentence: UOs occur frequently in SAB cases.  

INTRODUCTION:

Line #38 – Delete the extra space after ~ 20/100,000. 

Line #40 – Revise to eliminate extra terms.  SAB can be further classified as either complicated or uncomplicated infection.

Line #41 – Line #43 - Just to make it a little abbreviated please consider revising to: Adding further complexity to its occurrence, SAB may be referred to as community-acquired (CA) or health -care (HC) associated.  Such complexity makes clinical management challenging.

Line #46 – Change ½ to “nearly one-half of all” …..

Line #47 – Please consider revising to: patients suspected of SAB are routinely empirically….

Line #52 – Here just “lack of evidence-based……:” may be better.

Line #59 – Delete “in the course of SAB”.    

METHODS:

Line #62 – Line #63 – To avoid using “because” at the beginning of the paragraph, consider this minor revision.  This study used fully de-identified administrative data; therefore, it was exempt from ethics review under…..

Line #64 – Sinc in the first sentence you mention it was exempt from ethics review, not sure why you mention again “no ethics review was sought dure to……”.

Line #68 – Line #69 - Should “proven” be “documented”?  Then go on to say: we included MRSA and MSSA infections as well as complicated and uncomplicated cases.  This eliminates repeated “SAB”.

Line #70 – Above you referred to it as community-acquired, here you change it to community-onset; keep it consistent.

Line #78 – You abbreviated blood culture in the abstract, so use “BC” here.

Line #79 – May consider listing “IV” just after intravenous since you have used this abbreviation multiple times later in the text.

Line #80 – What do you mean “only the first SAB episode was included?  Is this during the study time frame, please clarify.

Line #94 – Change “principle” to “primary”

Line #127 – Change “of” to “or” nafcillin.

Line #128 – What do you mean by “all other was termed IET”, is these antibiotic choices????

Line #136 – “UO” has already been defined and abbreviated, so just use the abbreviation.

Line #148 – Should you say “ICU transfer following SAB onset”?

Line #152 – What is “SA” please clarify, I am assuming it is S. aureus.  Please spell out.

Line #154 – LOS has already been used and abbreviated multiple times, so just use LOS.   Then, delete the first LOS inside the ().

Line #157 – The numbering here seems a little odd since you are moving into secondary outcomes but continuing the numbers.  Maybe you should restart numbering with 1, 2, 3, and 4 for the secondary outcomes.

Line #159 – Revise to: post-infection onset hospital LOS (days).

Line #161 – MV has been defined and abbreviated, so just use abbreviated version “MV”.

Line #169 – Although it is a well know terminology, still consider placing (SD) after standard deviation since you have used SD abbreviation later.

Line #174 – Should you have a unit for the cell count? i.e., CFUs (colony-forming units!!!)

Line #175 – Abbreviate UOs.

Line #176 – Define AUROC (Area Under the Receiver Operating Characteristic Curve). 

RESULTS:

Line #180 – Revise to: Between 2020 and 2022, 4,080 patients…. Delete “years”, not needed.

Line #180 – Line #187 – The repetitive UOs and FOs makes this section little hard to follow.  Not sure if it can be revised to eliminate few terms.  Here is a suggestion starting Line #181: When comparing the groups, patients with UOs were more likely to be non-white, had multiple comorbidities, and were admitted to urban or teaching hospitals.  Finish this section with: Likewise, both groups……

Line #191 – Line #201 – Same as above, UO, FO, MV over and over and hard to follow, plus you just mentioned above that patients with UOs were sicker (multiple comorbidities).  Here is a suggestion:  During hospitalization and prior to the onset of SAB, those with UOs suffered from acute illnesses such as early septic shock, required ICU care, MV, vasopressor support and dialysis.  Here since the rest of the paragraph is discussing MV, you may revise to: More subjects in the UO group required MV later in the hospitalization; however, the mean SD duration in the UO group [] was similar to those in the FO group [].  In addition, septic shock occurred more frequently (~ 5 times) in the UO group.

Line #209 – Delete “of this” and just say: with the two most common measures being persistent bacteremia and secondary BSI.

Line #214 – This minor revision will eliminate few unnecessary terms: Although IET was more common in FO group than UO (), the rates were low in both groups.

Line #223 – Not sure if “prevalent” is a good term here.  I think “common’ may be better.

Line #224 – Please correct the location of the and closure of the parenthesis? I think this is what you are trying to report?  ….escalation (37.7%), more frequently a switch from vancomycin to daptomycin (Supplemental Material, File #3), …..

 Line #225 – Line #226 – Another minor revision to eliminate “post-infection….: The group with UOs had longer post-infection onset hospital LOS and higher costs. 

Line #230 – Just say: and a complicated SAB (Table 4).   

DISCUSSION:

Line #242 – I think “a considerable number” may fit better here than “a substantial proportion”.

Line #243 – Here now you use a different term “untoward outcome”, why not just use unfavorable outcome “UO“ as you have used throughout the text. 

Line #245 – Not sure it is appropriate to start a sentence with “and”.  So just start the sentence with: Across the entire cohort, nearly 1 in 4 survivors ()…  No need to repeat UO and FO since you refer to entire cohort, by now your readers should know this.

Line #251 – Delete the second “overall re-admission rate”, just say identical to our reported rate across …

Line #253 – Revise to: in contrast to our cohort where mortality was much lower at 5.1% among UOs and a mere 3% among the entire cohort.

 Line #261 – Abbreviate SAB, change “attendant” to “associated” or “related”.

Line #261 – Line #263 – Revise to: a recent systematic review (2011) investigating SAB related mortality reported one-month or later mortality rate ranged from 6% to 35%, with the mean of 20%.  Not sure what do you mean by “or later” at the end of this line!

Line #263 – Change “nearer” to “closer” for better terminology here.

Line #270 – Abbreviate UOs.  Just revise the sentence to: Other common UOs among patients with SAB included antimicrobial escalation in 1/3 during hospitalization; 1/3 required some period of vasopressor support, while 1 in 10 required dialysis or ICU admission. 

Line #277 – Not sure if you need to say: patients at risk for them.  I think if you just say “at risk patients” or “patients at risk” your readers will understand the concept.

Line #280 – Line #282 - A minor comment.  Delete “have”, just say we identified several.   Then, revise to: while having multiple comorbidities increases the risk of an UO significantly, the strongest predictors were related to the infection characteristics and treatment pattern. 

Line #285 – Line #290 – I am not sure if I am clear with this paragraph.  Please explain/clarify.

Line #297 – Consider changing “broad” to “extensive literature review”.  Here again you have “untoward outcome” vs. unfavorable outcome”!!!

Line #304 – You mention Group A, I did not see further grouping!!  Please clarify.

Line #314 – “Swath” vs. “scale”???   Abbreviate UO

Line #315 – Line #317 – …in patients ≥ 85 years of age aggressive interventions such as dialysis, ICU admission, MV and readmission may not be an optimal option.

Line #320 – A minor revision – One important aspect of our study is the inclusion of data from the beginning of COVID-19 pandemic in the US.

Line #341 – Abbreviate SAB.  

TABLES:

Tables 1, 2 and 3 are too long with compact but important information.     

Table #4 – Change “deficiency anemia” to just “anemias”, You have used two different definitions of MSSA: methicillin-susceptible and methicillin-sensitive (in the abbreviation list).   

SUPPLEMENTAL FILES:

A minor comment: Files 1 and 2 do not have (#) listed like File #3.  Either add or delete (#) for 3.

File #1 – Title MRSA and MSSA – You only define MRSA, also, to avoid too many abbreviations, consider spell out SA, it should not be difficult, I am assuming it is S. aureus.  You had this on line #152 as well.

Uncomplicated and Complicated - You only extensively define complicated.

For the last bullet point, please consider revising like:

CA – Within 3-days of admission

HA – On day 4 of admission or later

File #2 – I am not clear why this is provided since I did not see this term “Elix Hauser comorbidities” during my review.  Please clarify, I would suggest to just mention some of the most common comorbidities shared among the patients with UOs.  I am assuming patients did not have all these comorbidities.  

File #3 – No comment

Author Response

Appreciate the opportunity to review this manuscript on SAB and unfavorable outcomes drawn from a large sample out of a large database.  I enjoyed reading the manuscript.  Appreciate the supplemental material, tables and the figure which I am sure required extensive work.  As I read the manuscript, I found some areas in which clarification will be necessary and could further strengthen you work.  The English language and terminology will require minor editing.  Few inconsistencies among abbreviations.  I hope my comments will be well received and easy to follow. 

Good luck

--------------

MANUSCRIPT ID: antibiotics-3497311-v1

TITLE: Unfavorable outcomes and their risk factors in hospitalized patients with Staphylococcus aureus bacteremia in the US: A multicenter retrospective cohort study, 2020 -2022

GENERL COMMENTS: Thank you for your efforts on preparing this interesting manuscript.  I really enjoyed reading and even learning/refreshing a few terms during my review.  The manuscript is well written and for most parts flows well.  However, due to the multiple/many abbreviations and the repetitive nature (i.e., SAB, UO, FO), it was difficult to follow at some sections.  SAB is an infection, so at some parts you may just refer to it as such.  Few terminologies although correct, do not necessarily fit in the context and I will suggest revising them for better flow.  Multiple inconsistencies with the abbreviations, UO, FO, SAB, etc.   For most part, I pointed them out for you in specific lines.

AU: Thank you 

You have used “Unfavorable outcomes” (UOs) and untoward outcomes.  I think you should keep the terminology consistent.

AU: See below 

Not sure if you need to have the time frame (2020 – 2022) listed in the title.  Title is already long.  In addition, may consider reversing the order of the title: Risk factors associated with UOs in hospitalized patients with SAB in the US: A multicenter retrospective cohort study.

AU: Our preference is to keep the time frame.

ABSTRACT:

Line #16 – Since you have outcomes, the (UO) should be (UOs).  In addition, I think “incidence” or “frequency” may fit better than “prevalence”.

AU: Done

Line #17 – Not sure if your time frame 2020 – 2022 belongs here since we are discussing centers.  It can easily go to the end of the sentence (i.e., for S. aureus between 2020- 2022).

AU: Done, thanks.

Line #18 – you have 1 + BC positive for S. aureus.  Are you trying to say with one positive S. aureus blood culture?  Please clarify you have (+) and positive.  Do you mean more than one positive culture?

AU: We have changed to “at least one,” thanks.

Line #20 – Should you abbreviate (LOS) here since you have other abbreviations in the abstract.

AU: Done  

Line #21 – Add “and” before disease worsening.  Complete the term before abbreviating.  Favorable outcomes (FOs).  Then, we compared the UO group to favorable outcome (FO) group.

AU: Done 

Line #22 – I think “experienced” or “encountered” may be a better term than “suffered”.

AU: Replaced with “experienced.”

Line #25 – May consider changing to: required ICU admission….

AU: Done

Line #30 – Just a minor change in this short sentence: UOs occur frequently in SAB cases. 

AU: Changed to “in the setting of SAB.” 

INTRODUCTION:

Line #38 – Delete the extra space after ~ 20/100,000.

AU: Done 

Line #40 – Revise to eliminate extra terms.  SAB can be further classified as either complicated or uncomplicated infection.

AU: Apologies, but it is not clear what the reviewer would like us to remove.

Line #41 – Line #43 - Just to make it a little abbreviated please consider revising to: Adding further complexity to its occurrence, SAB may be referred to as community-acquired (CA) or health -care (HC) associated.  Such complexity makes clinical management challenging.

AU: Thank you. We will defer to the Journal on their style issues.

Line #46 – Change ½ to “nearly one-half of all” …..

AU: As above – will defer to the Journal’s style.

Line #47 – Please consider revising to: patients suspected of SAB are routinely empirically….

AU: We do not believe that patients can be suspected of SAB, but rather of suffering from it. Will leave as is for this reason.

Line #52 – Here just “lack of evidence-based……:” may be better.

AU: Again, this is a style issue.

Line #59 – Delete “in the course of SAB”.

AU: Can we ask the reviewer to clarify the reason for this request?    

METHODS:

Line #62 – Line #63 – To avoid using “because” at the beginning of the paragraph, consider this minor revision.  This study used fully de-identified administrative data; therefore, it was exempt from ethics review under…..

AU: Again, issue of style

Line #64 – Sinc in the first sentence you mention it was exempt from ethics review, not sure why you mention again “no ethics review was sought dure to……”.

AU: This is to clarify that we followed the guidance

Line #68 – Line #69 - Should “proven” be “documented”?  Then go on to say: we included MRSA and MSSA infections as well as complicated and uncomplicated cases.  This eliminates repeated “SAB”.

AU: Done

Line #70 – Above you referred to it as community-acquired, here you change it to community-onset; keep it consistent.

AU: Changed

Line #78 – You abbreviated blood culture in the abstract, so use “BC” here.

AU: Defer to Journal style

Line #79 – May consider listing “IV” just after intravenous since you have used this abbreviation multiple times later in the text.

AU: Done

Line #80 – What do you mean “only the first SAB episode was included?  Is this during the study time frame, please clarify.

AU: Yes, and we have now included the clarification

Line #94 – Change “principle” to “primary”

AU: “Principal” is the correct word

Line #127 – Change “of” to “or” nafcillin.

AU: Corrected, thank you

Line #128 – What do you mean by “all other was termed IET”, is these antibiotic choices????

AU: Correct. Have now clarified, thanks.

Line #136 – “UO” has already been defined and abbreviated, so just use the abbreviation.

AU: In the abstract, but not in the text

Line #148 – Should you say “ICU transfer following SAB onset”?

AU: Will defer to journal style

Line #152 – What is “SA” please clarify, I am assuming it is S. aureus.  Please spell out.

AU: Done

Line #154 – LOS has already been used and abbreviated multiple times, so just use LOS.   Then, delete the first LOS inside the ().

AU: This is the first mention in the text. But did alter in line 159 as requested.

Line #157 – The numbering here seems a little odd since you are moving into secondary outcomes but continuing the numbers.  Maybe you should restart numbering with 1, 2, 3, and 4 for the secondary outcomes.

AU: Done, thank you.

Line #159 – Revise to: post-infection onset hospital LOS (days).

AU: Done

Line #161 – MV has been defined and abbreviated, so just use abbreviated version “MV”.

AU: Done

Line #169 – Although it is a well know terminology, still consider placing (SD) after standard deviation since you have used SD abbreviation later.

AU: Done

Line #174 – Should you have a unit for the cell count? i.e., CFUs (colony-forming units!!!)

AU: This statistical “cell unit” terminology refers to the number of cases being compared, not the CFUs.

Line #175 – Abbreviate UOs.

AU: done

Line #176 – Define AUROC (Area Under the Receiver Operating Characteristic Curve).

AU: Done 

RESULTS:

Line #180 – Revise to: Between 2020 and 2022, 4,080 patients…. Delete “years”, not needed.

AU: Defer to Journal style

Line #180 – Line #187 – The repetitive UOs and FOs makes this section little hard to follow.  Not sure if it can be revised to eliminate few terms.  Here is a suggestion starting Line #181: When comparing the groups, patients with UOs were more likely to be non-white, had multiple comorbidities, and were admitted to urban or teaching hospitals.  Finish this section with: Likewise, both groups……

AU: We would appreciate details of what is unclear about this paragraph. In a cohort study where we compare two groups referring to those groups to present the results is unavoidable.

Line #191 – Line #201 – Same as above, UO, FO, MV over and over and hard to follow, plus you just mentioned above that patients with UOs were sicker (multiple comorbidities).  Here is a suggestion:  During hospitalization and prior to the onset of SAB, those with UOs suffered from acute illnesses such as early septic shock, required ICU care, MV, vasopressor support and dialysis.  Here since the rest of the paragraph is discussing MV, you may revise to: More subjects in the UO group required MV later in the hospitalization; however, the mean SD duration in the UO group [] was similar to those in the FO group [].  In addition, septic shock occurred more frequently (~ 5 times) in the UO group.

AU: Again, not sure how this is different or clearer. Thank you for this suggestion.

Line #209 – Delete “of this” and just say: with the two most common measures being persistent bacteremia and secondary BSI.

AU: OK

Line #214 – This minor revision will eliminate few unnecessary terms: Although IET was more common in FO group than UO (), the rates were low in both groups.

AU: Thank you. Again, stylistic, so will defer to Journal.

Line #223 – Not sure if “prevalent” is a good term here.  I think “common’ may be better.

AU: We would ask the reviewer to explain their reasoning for this statement. In fact, “common” is a synonym for “prevalent.”

Line #224 – Please correct the location of the and closure of the parenthesis? I think this is what you are trying to report?  ….escalation (37.7%), more frequently a switch from vancomycin to daptomycin (Supplemental Material, File #3), …..

AU: Thank you for this observation. However, we meant to place the parenthesis where it is, as the content within it refers to the percentage presented.

 Line #225 – Line #226 – Another minor revision to eliminate “post-infection….: The group with UOs had longer post-infection onset hospital LOS and higher costs.

AU: Again, we felt it was important to be clear that both the LOS and costs were those following the onset of infection.  

Line #230 – Just say: and a complicated SAB (Table 4).

AU: Here it was important to stress that it was, in fact, only a complicated SAB.c   

DISCUSSION:

Line #242 – I think “a considerable number” may fit better here than “a substantial proportion”.

AU: Can the reviewer give us a rationale for this? We feel that we did not mis-state the result when referring to 60% of the cohort.

Line #243 – Here now you use a different term “untoward outcome”, why not just use unfavorable outcome “UO“ as you have used throughout the text.

AU: OK 

Line #245 – Not sure it is appropriate to start a sentence with “and”.  So just start the sentence with: Across the entire cohort, nearly 1 in 4 survivors ()…  No need to repeat UO and FO since you refer to entire cohort, by now your readers should know this.

AU: Thank you. We prefer to keep the wording as is for clarity.

Line #251 – Delete the second “overall re-admission rate”, just say identical to our reported rate across …

AU: Again, included for maximal clarity.

Line #253 – Revise to: in contrast to our cohort where mortality was much lower at 5.1% among UOs and a mere 3% among the entire cohort.

AU: Stylistic, as above.

 Line #261 – Abbreviate SAB, change “attendant” to “associated” or “related”.

AU: Will defer to Journal style

Line #261 – Line #263 – Revise to: a recent systematic review (2011) investigating SAB related mortality reported one-month or later mortality rate ranged from 6% to 35%, with the mean of 20%.  Not sure what do you mean by “or later” at the end of this line!

AU: “Later” refers to later than 2011.

Line #263 – Change “nearer” to “closer” for better terminology here.

AU: Stylistic

Line #270 – Abbreviate UOs.  Just revise the sentence to: Other common UOs among patients with SAB included antimicrobial escalation in 1/3 during hospitalization; 1/3 required some period of vasopressor support, while 1 in 10 required dialysis or ICU admission.

AU: The first sentence in the paragraph is a transition sentence.  

Line #277 – Not sure if you need to say: patients at risk for them.  I think if you just say “at risk patients” or “patients at risk” your readers will understand the concept.

AU: Included for maximal clarity.

Line #280 – Line #282 - A minor comment.  Delete “have”, just say we identified several.   Then, revise to: while having multiple comorbidities increases the risk of an UO significantly, the strongest predictors were related to the infection characteristics and treatment pattern. 

AU: With respect, this is again a stylistic request.

Line #285 – Line #290 – I am not sure if I am clear with this paragraph.  Please explain/clarify.

AU: Can the reviewer please be more specific what requires clarification? We present a standard discussion of possible confounding by indication as a possibility for our findings. We also acknowledge that it may be a true relationship, which would imply that current therapeutic approaches fall short.

Line #297 – Consider changing “broad” to “extensive literature review”.  Here again you have “untoward outcome” vs. unfavorable outcome”!!!

AU: Changed to unfavorable

Line #304 – You mention Group A, I did not see further grouping!!  Please clarify.

AU: This refers to citation #4.

Line #314 – “Swath” vs. “scale”???   Abbreviate UO

AU: This is correct

Line #315 – Line #317 – …in patients ≥ 85 years of age aggressive interventions such as dialysis, ICU admission, MV and readmission may not be an optimal option.

AU: Not sure what the reviewer would like us to address. Is our statement incorrect?

Line #320 – A minor revision – One important aspect of our study is the inclusion of data from the beginning of COVID-19 pandemic in the US.

AU: Again, did we mis-state something?

Line #341 – Abbreviate SAB.

AU: We prefer to leave as is.  

TABLES:

Tables 1, 2 and 3 are too long with compact but important information.

AU: Agree info is important     

Table #4 – Change “deficiency anemia” to just “anemias”, You have used two different definitions of MSSA: methicillin-susceptible and methicillin-sensitive (in the abbreviation list).

AU: This is the accurate description of the coding algorithm.    

SUPPLEMENTAL FILES:

A minor comment: Files 1 and 2 do not have (#) listed like File #3.  Either add or delete (#) for 3.

AU: Do not see what the reviewer is referring to

File #1 – Title MRSA and MSSA – You only define MRSA, also, to avoid too many abbreviations, consider spell out SA, it should not be difficult, I am assuming it is S. aureus.  You had this on line #152 as well.

AU: MSSA is any SA that is not MRSA. The abbreviations are widely used and do not require spelling out the way they do in the text.

Uncomplicated and Complicated - You only extensively define complicated.

AU” Uncomplicated is all that is not complicated

For the last bullet point, please consider revising like:

CA – Within 3-days of admission

HA – On day 4 of admission or later

AU: Thank you. Again, question of style

File #2 – I am not clear why this is provided since I did not see this term “Elix Hauser comorbidities” during my review.  Please clarify, I would suggest to just mention some of the most common comorbidities shared among the patients with UOs.  I am assuming patients did not have all these comorbidities.

AU: Line 183: “Supplemental File 2 for individual comorbidities” 

“Elixhauser comorbidities” are a well know system to identify them in administrative data.  

File #3 – No comment

AU: OK

Reviewer 4 Report

Comments and Suggestions for Authors

This study compared several clinical variables between patients with staphylococcus aureus bacteremia. The text is well written, it is easy to read and understand. The authors could look into the following comments to improve the paper:

Comments:

(1) In the abstract. Please add the the values when describing the data and showing the comparisons between UOS and the unfavorable group.

(2) Please add the p values when describing the odd ratios. Alternatively, please add "all p values < 0.05#.

(3) In the analysis, was the data stratified according to patients with MRSA and non-MRSA?

(4) Could you please explain why the 3 days after admission is the cutoff between community and nosocomial infection?

(5) Regarding the data source (section 2.3). Could you please add this data as supplementary so it is possible to reproduce the several analyses?

(6) Sorry to ask but, what is the purpose of AUROC and H-L goodness-of-fit test?

(7) In the Tables, the disease of the patient is not mentioned. Should this variable be included? A different pathological condition of the patient may affect the evolution and response to treatment.

(8) Age >80 years associated with less provability of unfavorable outcome (odds ratio 0.687). Is this logical?

(9) In Table 4. Could you please calculate again using binary logistic regression, but with backwald conditional?

(10) Was the type of treatment for SAureus comparable between patients? Could the type of antibiotic have an effect?

(11) Instead of gmail accounts, could you please use Institutional emails?

Author Response

This study compared several clinical variables between patients with staphylococcus aureus bacteremia. The text is well written, it is easy to read and understand. The authors could look into the following comments to improve the paper:

Comments:

  • In the abstract. Please add the the values when describing the data and showing the comparisons between UOS and the unfavorable group.

AU: The numbers are in the Results section. We are frankly puzzled by this request. If the reviewer could be more specific, we would be grateful.

  • Please add the p values when describing the odd ratios. Alternatively, please add "all p values < 0.05#.

AU: The 95% confidence intervals provided obviate the need for presenting p values.

  • In the analysis, was the data stratified according to patients with MRSA and non-MRSA?

AU: We did not perform this stratified analysis. We did include it in the regression, as shown in Table 4.

(4) Could you please explain why the 3 days after admission is the cutoff between community and nosocomial infection?

AU: The literature suggests that 48-72 hours is the window for the CAI threshold. In the Premier database, time is given as days. In all of our studies we have used 3 days as the cut-off.

(5) Regarding the data source (section 2.3). Could you please add this data as supplementary so it is possible to reproduce the several analyses?

AU; Unfortunately, that is not possible, as we stated in the data availability statement:

Data Availability Statement: The dataset analyzed in the current study was used under a license from PINC AI, and is available from the corresponding author on reasonable request and with permission from PINC AI.”

(6) Sorry to ask but, what is the purpose of AUROC and H-L goodness-of-fit test?

AU: AUROC tests for discrimination and H-L GoF for calibration of the model.

(7) In the Tables, the disease of the patient is not mentioned. Should this variable be included? A different pathological condition of the patient may affect the evolution and response to treatment.

AU: As shown in Table 3, over 80% of all BSIs were primary.

(8) Age >80 years associated with less provability of unfavorable outcome (odds ratio 0.687). Is this logical?

AU: We discuss this finding at length in the Discussion section para starting in line #309. At the very end of that paragraph we present an additional conjecture:

“One hypothesis for this finding may be that patients 85 years and older may not be subject to such aggressive interventions as dialysis, ICU, MV, or readmission. In this way, this lack of aggressive treatment we defined as UO may be masking withdrawal of aggressive care rather than a true FO. Future studies should address this apparent paradox in further detail.”

(9) In Table 4. Could you please calculate again using binary logistic regression, but with backwald conditional?

AU: Firstly, we cannot alter the protocol that was developed prior to the conduct of the analyses, as that would introduce a bias into our results. Secondly, it is unclear to us why the reviewer is requesting this.

(10) Was the type of treatment for SAureus comparable between patients? Could the type of antibiotic have an effect?

AU: Table 3 lists the antibiotics. Table 4 lists the predictors, including vancomycin and daptomycin.

(11) Instead of gmail accounts, could you please use Institutional emails?

AU: These are the emails we have used for nearly two decades with no issues.